# Is Pain Perception Communicated through Mothers? Maternal Pain Catastrophizing Scores Are Associated with Children’s Postoperative Circumcision Pain

**DOI:** 10.3390/jcm12196187

**Published:** 2023-09-25

**Authors:** Sevda Akdeniz, Ahmet Haydar Pece, Hatice Selcuk Kusderci, Serkan Dogru, Serkan Tulgar, Mustafa Suren, Ismail Okan

**Affiliations:** 1Department of Anesthesiology and Reanimation, Samsun Training and Research Hospital, Samsun University, 55090 Samsun, Turkey; ahmet5396@hotmail.com (A.H.P.); drkusderci@hotmail.com (H.S.K.); serkantulgar.md@gmail.com (S.T.); drmustafasuren@gmail.com (M.S.); 2Department of Anesthesiology and Reanimation, Mersin City Hospital, 33240 Mersin, Turkey; drserkandogru@gmail.com; 3Department of General Surgery, Faculty of Medicine, Istanbul Medeniyet University, 34720 Istanbul, Turkey; hismailok@yahoo.com

**Keywords:** children, mother, pain catastrophizing, postoperative pain, circumcision, FPS, VAS

## Abstract

The aim of this study was to evaluate the relation of maternal pain catastrophizing score with children who underwent circumcision postoperative pain. This prospective cohort study was performed between March 2022 and March 2023 at Samsun University, Turkey. Demographic characteristics of mothers and children, mothers’ education level, presence of chronic pain, and Beck Depression Inventory scores were recorded preoperatively. Pain catastrophizing was assessed by applying the pain catastrophizing scale (PCS) to the mothers of children who experienced postoperative circumcision pain. The mothers were divided into low-pain catastrophizing (Group 1) and high-pain catastrophizing (Group 2) group. A total of 197 mothers and sons participated in the study, with 86 (43.6%) in Group 1 and 111 (56.4%) in Group 2. Significant differences were found between the two groups in terms of the mothers’ PCS scores (*p* < 0.001), education levels (*p* = 0.004), chronic pain scores (*p* = 0.022), and Beck Depression Inventory scores (*p* < 0.001). Our findings showed that children with high pain catastrophizing mothers experience greater postoperative pain than those with low pain catastrophizing mothers. This may be attributable to a mother’s specific cognitive style for coping with pain, which is associated with the child’s responses to painful experiences.

## 1. Introduction

Pain is a sensation experienced by all human beings. However, individual reactions to the same wounds or pain-inflicting situations differ depending on the social context [1]. Many biological, psychological, and socio-cultural factors have been identified as components affecting the experience of pain. The manner in which parents share memories of painful events from their pasts with their children has been associated with children’s cognitive processes (e.g., memory) and broader socio-emotional development [2]. Children often develop their pain perceptions by observing pain in others. Empathy towards pain is essential for noticing, understanding, and responding to it in others. Children develop empathy toward pain gradually, which may be related to social learning. Feeling empathy toward another’s pain is a multidimensional biological, emotional, and behavioral response that allows children to notice that someone is in pain, to understand the experience of pain, and to respond in a prosocial manner [3]. The family plays an important role in children’s displays of pain-related behaviors within a predefined sociocultural context [4]. Previous research has shown a link between pain-related behaviors and pain perception in parents and their offspring [5]. The children of parents with severe illnesses have been reported to register higher symptom scores than a control group, representing evidence for a family-based pain model [6]. Pain catastrophizing may be an exaggerated negative response to actual or anticipated pain and pain-related behavior that entails greater feelings of helplessness than the normal population in terms of the pain experienced [7]. Such catastrophizing has been identified as a significant determinant of the intrapersonal components of pain, including heightened pain intensity, discomfort, and disability. The phenomenon has also been regarded as a social communicative role in a broader, interpersonal, or societal coping style [8]. Children with chronic illnesses exhibit divergent responses to experimental pain stimuli and employ distinct coping strategies compared to their healthy counterparts. Furthermore, studies have revealed a marked similarity between the pain-coping methods adopted by these children and those employed by their parents [9].

Circumcision is an important surgical procedure in Turkey. Despite being purely surgical, it also has a rich cultural significance. Boys are prepared physically and mentally for this landmark event. Turkish culture attributes a very powerful significance to circumcision, representing the passage from boyhood to adulthood. The extended family and even the entire community participate in the ceremony. The procedure is also regarded as a religious observance. Boys are forewarned of the pain of circumcision by their parents and friends. However, they are frequently informed that they should not show pain even if they feel pain. Pain has, therefore, already been considered by the boys themselves and the wider culture by the time the procedure takes place. From another perspective, circumcision pain gives researchers a unique opportunity to investigate the experience of pain in the context of family and traditional culture.

This study investigated the relationship between maternal pain catastrophizing score levels and the postoperative circumcision pain experienced by children. 

## 2. Material and Methods

### 2.1. Study Design 

The research was designed as an observational cross-sectional study. One hundred ninety-seven mothers and their sons who underwent circumcision at the Samsun University, Samsun Training and Research Hospital, Samsun, Turkey, between March 2022 and March 2023 were included in the study. The children were recruited consecutively. Boys who presented to the preoperative anesthesia outpatient clinic with their mothers for elective circumcision were enrolled if their mothers agreed to participate. These mothers at the anesthesia outpatient clinic were given a questionnaire to complete and provided informed consent. Children whose mothers were not present were not included. The study was conducted when the same anesthesiologist was in the outpatient clinic. All these procedures were performed outside the outpatient clinic, and the mothers were asked to bring these forms with them on the day of surgery.

#### 2.1.1. Inclusion Criteria

-Age 5–12 years, and-Being operated on under general anesthesia.

#### 2.1.2. Exclusion Criteria

-Children attending without their biological mothers or with a parent other than the mother,-Children aged under five or over 12,-History of previous surgery,-Being operated under regional anesthesia,-Use of analgesic, antiepileptic, or sedative medications, and-Other procedures are being performed in addition to circumcision (such as herniorrhaphy, tonsillectomy, orchiopexy, and appendectomy).

Two hundred twenty-eight patients met the inclusion criteria during the research. However, 19 were subsequently excluded on account of unwillingness to participate, and 12 due to being lost to follow-up (Figure 1).

### 2.2. The Procedure

After the children had been taken to the operating room, general anesthesia was induced with intravenous 2–3 mg/kg propofol, after which a dorsal penile block using 0.2 mL/kg of 0.5% bupivacaine was applied by the pediatric surgeon. Maintenance anesthesia consisted of 1 MAC sevoflurane in a 50% oxygen-nitrous oxide mixture. The dorsal slit circumcision method was performed on all children by the same surgeons.

All the children included in the study received intravenous acetaminophen (15 mg per kg body weight) for pain control.

The mothers participating in the study were evaluated using the pain catastrophizing scale (PCS) and the Beck Depression Inventory.

#### 2.2.1. Pain Catastrophizing Scale

The PCS is a 13-item self-report inventory developed in 1995 and used to determine how much individuals catastrophize in response to pain [10]. Total possible scores range from 0 to 52, with higher scores indicating negative results. 

The Turkish-language version has been tested for reliability and validity, and the Cronbach alpha coefficient was measured at 0.90. PCS scores of 16 or lower are regarded as normal, and scores of 17 or above are high [11]. In the present study, PCS scores of 16 or less were regarded as low, and these mothers were assigned to Group 1. PCS scores of 17 or more were regarded as high; these mothers constituted Group 2.

#### 2.2.2. Beck Depression Inventory

The Beck Depression Inventory, consisting of 21 items scored from 0 to 3, was used to assess psychological status. Each item evaluates a depression-associated symptom during the previous two weeks. Based on the scores achieved, each patient’s depressive state was calculated as minimal (0–13 points), mild (14–19 points), moderate (20–28 points), or severe (29–63 points) [12].

The mothers were divided into a low pain catastrophizing group (Group 1) and a high pain catastrophizing group (Group 2) based on their PCS scores [11].

### 2.3. Evaluation of Postoperative Pain in Children

Children’s pain was evaluated by means of a visual analog scale (VAS) and the faces pain scale (FPS). Scores were recorded one- and three hours post-circumcision.

#### 2.3.1. Visual Analog Scale 

The VAS instructs patients to mark a point on a defined scale to indicate their pain intensity. The scale can measure the worst, least, or average pain over the postoperative period. While the VAS is quick to use, it also requires good visual acuity, dexterity, and the use of pen and paper. The pain severity VAS has been shown to exhibit excellent reliability and validity [13].

#### 2.3.2. Faces Pain Scale

The FPS consists of six drawn faces ranging from ‘no pain’ to ‘unbearable pain’. The faces indicative of pain appear happy at one end of the spectrum to crying at the other. Each facial expression represents a score of 0, 2, 4, 6, 8, or 10, and the patient’s pain is matched to the appropriate face on the scale. The FPS is a valid, reliable, and popular pain measurement tool widely employed to evaluate the intensity of acute postoperative pain in children [13].

All scores for both mothers and children were determined by the same anesthetist (A.H.P.).

### 2.4. Statistical Analysis

The data were analyzed on SPSS version 25 software (Statistical Package for Social Sciences-IBM Corp., Armonk, NY, USA). A *p* values < 0.05 was considered significant. Nominal data were expressed as frequency and percentage values. Normally distributed continuous data were presented as mean ± standard deviation, and non-normally distributed continuous data as median values (interquartile range [IQR]: 25th percentile–75th percentile). The normality of the distribution of continuous variables was calculated using the Kolmogorov-Smirnov and Shapiro-Wilk tests. The Independent Samples T test was applied to evaluate whether normally distributed continuous variables differed significantly between the two groups. The Mann-Whitney U test was applied to determine whether differences between the two groups in terms of non-normally distributed continuous variables were statistically significant. Nominal variables were analyzed using the Chi-square and Fisher’s exact tests.

## 3. Results

One hundred ninety-seven mothers and sons with mean ages of 35.43 ± 5.82 and 6.85 ± 1.27 years, respectively, were included in the study. 

Maternal age, employment, marital status, number of children, and presence of chronic disease were similar between the groups. However, significant differences were observed between the groups in terms of PCS scores (*p* < 0.001), education (*p* = 0.004), and Beck Depression Inventory scores (*p* < 0.001). Chronic pain was observed in 33.7% of the patients in Group 1 and 52.3% in Group 2, the difference being statistically significant (*p* = 0.009). VAS scores among patients with chronic pain in Group 1 were significantly lower than in Group 2 (*p* = 0.004). Maternal characteristics are presented in Table 1.

No statistically significant difference was found between the groups in terms of children’s ages, education, or presence of chronic disease. The sociodemographic features of the children are listed in Table 2. 

The groups’ VAS scores at one and three hours postoperatively were 1.82 ± 1.09 vs. 2.54 ± 1.5, respectively, in Group 1 and 0.98 ± 0.91 vs. 1.73 ± 1.29 in Group 2. Significant differences were observed between the groups’ mean postoperative VAS pain scores (*p* < 0.001). FPS scores also differed significantly between the groups at one and three hours after circumcision (2.37 ± 1.51 vs. 3.35 ± 1.81 and 1.02 ± 1.13 vs. 2.19 ± 1.17 respectively; *p* < 0.001 for both). The children’s groups’ detailed pain scores are shown in Figure 2.

## 4. Discussion

This study showed that the sons of mothers with high pain catastrophizing scores experienced significantly greater post-circumcision pain. In addition, mothers with higher PCS scores were more likely to suffer depression and chronic pain and have lower levels of education.

Psychosocial components play a significant role in modulating and determining the experience of pain. Painful behaviors can be learned, and as role models for their children, parents can be models for pain learning and coping strategies. A previous experimental study demonstrated that parents who had experienced pain influenced their children through verbal communication and interpersonal interactions, thus conveying their destructive thoughts. Those authors showed the impact of verbal communication on the catastrophization of experienced pain [5]. Publications have also examined the parent-child gender relationship and its impact on anxiety and pain. Researchers have stated that children whose parents exhibit exaggerated responses also tend to exhibit exaggerated anxiety. How parents express pain can also influence their children’s anxiety levels, and the anxiety status of children with parents who exhibit exaggerated pain responses is also gender-specific [14]. Studies have also shown a linear relationship between maternal modeling of fear and anxiety expression compared to paternal modeling, indicating that mothers’ behaviors can influence children’s pain perception and anxiety levels [15,16]. The present study demonstrated a linear relationship between high maternal PCS scores and children’s postoperative pain scores. This suggests that children may adopt their mothers’ reactions to pain as their coping mechanism and may exaggerate their pain experiences. Previous studies have indicated that parents’ responses to pain influence their children’s pain reactions and pain-coping behavior [17].

Circumcision, a traditional ritual, entails a powerful social component, and the behaviors of parents and the community toward postoperative pain differ from those involved in other surgeries. We, therefore, attempted to minimize stress factors by having mothers complete the PCS questionnaire in a calm environment before the operation took place, and the child’s postoperative pain and the mother’s preoperative PCS score were evaluated in association with this ritual. The impact of psychosocial components was also investigated. The children of mothers with higher pain catastrophizing scores experienced more postoperative pain in this study. In contrast to studies that have shown gender-specific relationships between parental presence and the presence of a pain model (usually a parent) and pain and psychological outcomes in children with chronic pain, the purpose of the present study was to bring a different perspective to the literature by investigating psychosocial interactions in a traditional surgical procedure [18].

Two previous studies considered postoperative pain risk factors after major surgery in children. They showed the relationship between parental and child pain catastrophization and the presence of pain catastrophization in children cared for by individuals with high anxiety. However, the principal limitation of both these studies was the lack of preoperative evaluations [19,20]. The present study is particularly valuable from that perspective, and the PCS scores of mothers examined in the preoperative outpatient clinic (without stress and in the absence of pain) are significant as a baseline datum. This is important because the PCS score, investigated in mothers in a calm period before the onset of pain in children, shows the effect of postoperative pain in children after circumcision surgery.

The presence of chronic pain and VAS scores were significantly higher in women with high pain catastrophization scores, 52.3% of whom had histories of chronic pain. A higher prevalence of chronic pain may be expected in patients with high pain catastrophization. However, studies have reported that pain catastrophization can develop even in a population without pain complaints [21]. A study comparing neuropathic and non-neuropathic patients said that only 30% of individuals with chronic pain due to neuropathy experienced catastrophic pain. However, the authors of that study suggested that pain catastrophization can also occur in painless neuropathic patients and healthy individuals [22]. Given this information, chronic pain may be considered an expected finding in approximately half of patients with high catastrophic pain scores. In addition, depression scores were significantly higher in patients with high pain catastrophization scores, and high rates of depression in patients with high catastrophic pain scores can be considered an expected finding.

Although our study results were consistent with the FPS and VAS scores in previous research in terms of postoperative pain levels in children [23,24,25,26], it is also important to consider the effect of maternal PCS scores, which we examined preoperatively, on children’s post-circumcision pain.

Circumcision in Turkish culture represents an important milestone in the child’s development, particularly in its emphasis on not feeling and showing pain. This study examined the effect of the mother’s preoperative PCS score on the child’s postoperative circumcision surgery pain. In contrast to other elective surgeries, circumcision was found to cause more postoperative pain in the children of mothers with high PCS scores. However, the procedure was performed entirely at the request of the family.

Several factors affect postsurgical pain in children, and presurgical risk factors that contribute to such pain were not fully evaluated in our study. Among those studies investigating postsurgical pain in children, one reported a relationship between preoperative baseline pain levels, child anxiety, pain self-efficacy and postsurgical pain [27]. Another observed an association between postsurgical pain and parental catastrophizing [28]. The common finding in both these studies was that age and sex were not related to postsurgical pain, and similar studies in the literature have also found no association between postsurgical pain and age or sex [19,29]. In addition, no relationship has been found between postsurgical pain and the child’s BMI, time since diagnosis, child negative mood, family income, or child pain catastrophizing [27,28].

No children in our study had any cognitive disabilities, although assessing pain in such children is very difficult. Children with such conditions, including intellectual, developmental, and learning disabilities and cognitive impairment deriving from acquired brain injuries or neurodegenerative diseases, are known to be more at risk of pain, with greater disability being associated with more severe pain [30]. Pain evaluation and management are problematic due to communication obstacles. Postoperative pain is not adequately assessed in these children, and such patients may receive lower amounts of analgesics and experience more pain throughout the entire perioperative period or in the treatment of postoperative pain than other pediatric groups [31,32,33]. Specific tools such as the revised Face, Legs, Activity, Cry and Consolability Scale, the Paediatric Pain Profile, the Pain Behaviour Checklist, and the Non-communicating Children’s Pain Checklist should be applied according to the child’s disease (such as autism spectrum disorder, Down syndrome, etc.) in these cases. However, due to the lack of a suitable measurement tool for this population, pain can often only be assessed by applying the most appropriate tool or a combination thereof on a case-by-case basis. Pain evaluation in such patients requires highly specific skills and knowledge. Medical professionals should receive special training for performing this, together with parents or other caregivers, using pain scales [33].

The principal limitations of this study include its single-center nature and the fact that fathers were not included. It should also be remembered that pain is a subjective experience that can vary depending on the mental and psychological state of the individual concerned. Another important limitation is that we did not assess postoperative recovery. This is a complex phenomenon affected by various factors, including patient characteristics, surgical procedures, and anesthesia [26]. The patient’s ability to continue normal activities following surgery and anesthesia is an important indicator of successful postoperative recovery [34]. Several questionnaires showing the quality of postoperative recovery are available. The Quality of Recovery (QoR) questionnaire is one of the tools used to assess the quality of recovery and health status in the early postoperative stages [35]. Although different versions exist, such as QoR-15 and QoR-40, both consist of questions evaluating patients’ pain, physical comfort, independence, psychological support, and emotional state. Higher results on the QoR assessment tool, which has been translated and is widely employed in several countries, including Türkiye, indicate a better quality of recovery [36,37].

Despite the above limitations, this study is the first to reveal the relationship between the pain experienced by children and maternal PCS scores following a surgical procedure with a very strong cultural background, such as circumcision. With its combination of cultural and familial factors, circumcision represents a unique model for developing pain perception in children.

## 5. Conclusions

Traditional circumcision surgery has a significant effect on children’s pain perceptions, depending on the mother’s level of pain catastrophization. Our results are consistent with the previous literature but are essential because this research is the first to investigate maternal catastrophic pain scores before surgery. However, a larger sample would have enhanced the generalizability of the results, and more research considering the roles of both parents is now needed. The results of this study show that the children of mothers with high levels of catastrophic pain before surgery experienced more postoperative pain than those of mothers with low levels of catastrophic pain. In addition, catastrophic pain and education levels contribute to our understanding of the relationship between chronic pain and depression. For all these reasons, it is important to evaluate maternal pain at a catastrophic level to plan appropriate approaches and measures for reducing the potential adverse effects on children’s postoperative pain. The experience of pain and catastrophization is affected by psychosocial variables, genetic factors, gender, and many other factors. An improved understanding of these numerous factors that affect patients’ experience of pain may assist in the provision of more targeted therapeutic options and reduce the likelihood of pain becoming chronic.

## Figures and Tables

**Figure 1 jcm-12-06187-f001:**
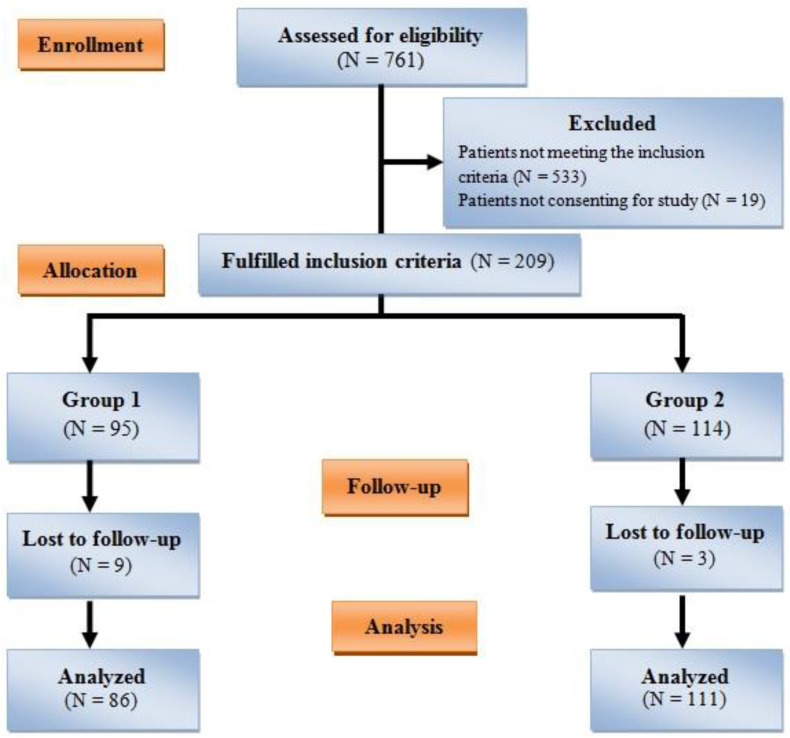
Patients’ enrollment algorithm.

**Figure 2 jcm-12-06187-f002:**
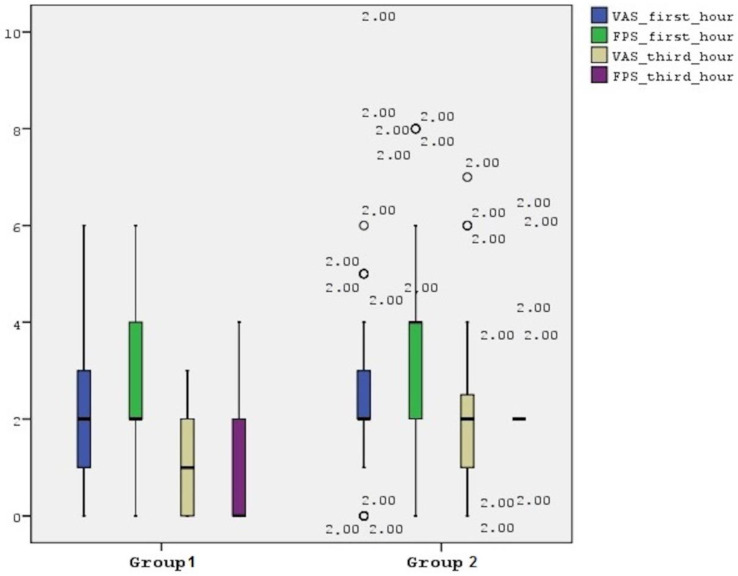
Children’s postoperative first and third-hour pain levels (FPS, faces pain scale; VAS, visual analog scale).

**Table 1 jcm-12-06187-t001:** Detailed information on the mothers in the study.

Variables	Group 1 (N = 86)	Group 2 (N = 111)	*p*
Age (years, mean ± SD)	35.09 ± 6.09	35.7 ± 5.62	0.468
PCS-TR score (mean ± SD)	8.26 ± 4.94	27.84 ± 8.64	<0.001
Education (N, %) Primary High school University	20 (23.2%) 43 (50%) 23 (26.8%)	54 (48.7%) 36 (32.4%) 21 (18.9%)	0.004
Employment (N, %) Housewife Service sector Nurse Teacher Officer	59 (68.6%) 13 (15.1%) 5 (5.8%) 5 (5.8%) 4 (4.7%)	84 (75.7%) 20 (18%) 2 (1.8%) 4 (3.6%) 1 (0.9%)	0.289
Marital status (N, %) Married Divorced	83 (96.5%) 3 (3.5%)	107 (96.4%) 4 (3.6%)	0.965
Number of children (mean ± SD)	2.22 ± 0.7	2.03 ± 0.72	0.075
Presence of chronic disease (N, %)	14 (16.3%)	20 (18%)	0.496
Presence of chronic pain (N, %)	29 (33.7%)	58 (52.3%)	0.009
VAS of chronic pain	3.03 ± 1.37	3.98 ± 1.44	0.004
Beck Depression Inventory score *	6 (3–10)	12 (7–18)	<0.001

* Data presentation of median and interquartile range (IQR, 25th–75th percentile). Abbreviations: PCS-TR, the Turkish version of pain catastrophizing scale; VAS, visual analog scale.

**Table 2 jcm-12-06187-t002:** Sociodemographic features of the children in the study.

Variables	Group 1 (N = 86)	Group 2 (N = 111)	*p*
Age (years, mean ± SD)	6.88 ± 1.3	6.82 ± 1.26	0.766
Education (N, %) Nursery Kindergarten Primary school	12 (14%) 19 (22.1%) 55 (62.9%)	11 (9.9%) 29 (26.1%) 71 (64%)	0.837
Presence of chronic disease (N, %)	9 (10.5%)	17 (15.3%)	0.594

## Data Availability

The data that support the findings of this study are available from the corresponding author, S.A., upon reasonable request.

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
