# Peer review of "Is Pain Perception Communicated through Mothers? Maternal Pain Catastrophizing Scores Are Associated with Children’s Postoperative Circumcision Pain"

_jcm, 2023, doi:10.3390/jcm12196187_

Round 1
Reviewer 1 Report
This paper deals with the maternal pain catastrophizing score in relation to postoperative pain in infants after circumcision.
The introduction, materials and methods are clearly presented and the results focus on the question asked.
Several suggestions to improve the manuscript:
- No scores are considered for experience of surgery or anaesthesia with a score as EVAN or rehabilitation as QORscore. It would be interesting to evaluate if the difference seen in the immediate postoperative phase is confirmed later or if it is only an episodic event with a difference in expression between the two groups. If the data are not available, please discuss this in the Discussion section of the manuscript.
- In the results section, it would be interesting to present Table 2 in a more graphical way as a box plot to better see the overlap between the two groups.
Author Response
We the authors are most grateful for your kind and valuable comments. Our answers to those comments appear beneath the relevant sentences. All the necessary changes have made in the manuscript in line with the reviewer’s suggestions. All changes have been highlighted in yellow in the text.
Several suggestions to improve the manuscript:
1. No scores are considered for experience of surgery or anaesthesia with a score as EVAN or rehabilitation as QOR score. It would be interesting to evaluate if the difference seen in the immediate postoperative phase is confirmed later or if it is only an episodic event with a difference in expression between the two groups. If the data are not available, please discuss this in the Discussion section of the manuscript.
Unfortunately, EVAN or QOR-15 scores were not evaluated in this study. We are therefore unable to provide information about these two questionnaires. We profoundly regret that we omitted to examine such an important parameter.We hope for your understanding on this matter. However, a discussion aroundQOR-15 scores has been given in the Discussion section.
2. In the results section, it would be interesting to present Table 2 in a more graphical way as a box plot to better see the overlap between the two groups.
In accordance with your suggestion, children’sage, education, and presence of chronic disease have been left in Table 2 while VAS and FPS values have been redesigned as a boxplot as Figure 1.
Reviewer 2 Report
I thought the concept was excellent, especially the use of preoperative collection of data from the mothers.
You mention that half of high PCS mothers had history of chronic pain. Did you rerun the data removing chronic pain from both groups? An interesting question: would simply high PCS lead to increased postoperative pain, because the presence of chronic pain between the 2 groups is also significant at 0.009. Is high PCS influenced by the history of chronic pain? It might be useful to try and separate out.
That is the one weakness I identify in that what has more influence...high PCS or presence of chronic pain. It also doesn't specify if the chronic pain aspect is current or historical and resolved.
It would be great to include this information in a second set of tables.
Only a couple points in the Abstract.
line 19...BDI scores were recorded preoperative...should be preoperatively.
The next sentence in lines 19-21 is awkward and should be worded more clearly.
I thought the English in the body of the text was good. It may be the abstract got cut and pasted in an odd fashion.
Author Response
We the authors are most grateful for your kind and valuable comments. Our answers to your comments appear under the sentences concerned. All the necessary changes have made in the manuscript in line with the reviewer’s suggestions. All changes have been highlighted in yellow in the text.
1. You mention that half of high PCS mothers had history of chronic pain. Did you rerun the data removing chronic pain from both groups? An interesting question: would simply high PCS lead to increased postoperative pain, because the presence of chronic pain between the 2 groups is also significant at 0.009. Is high PCS influenced by the history of chronic pain? It might be useful to try and separate out. That is the one weakness I identify in that what has more influence...high PCS or presence of chronic pain. It also doesn't specify if the chronic pain aspect is current or historical and resolved. It would be great to include this information in a second set of tables.
Since our study investigated pain levels in the children of mothers with high PCS, no detailed examination of chronic pain in mothers with such pain was performed. The information you request is therefore sadly unavailable.We profoundly regret having omitted to examine such an important parameter and hope for your understanding on this matter.
2. Only a couple points in the Abstract.
The requested amendments have been made to the Abstract section. We are additionally grateful to you for drawing our attention to these errors that escaped our notice.